# Emergence of Recombinant Subclade D3/Y in Coxsackievirus A6 Strains in Hand-Foot-and-Mouth Disease (HFMD) Outbreak in India, 2022

**DOI:** 10.3390/microorganisms12030490

**Published:** 2024-02-28

**Authors:** Sanjaykumar Tikute, Pratik Deshmukh, Nutan Chavan, Anita Shete, Pooja Shinde, Pragya Yadav, Mallika Lavania

**Affiliations:** 1Enteric Viruses Group, ICMR-National Institute of Virology, 20-A, Dr. Ambedkar Road, Pune 411001, Maharashtra, India; sanjaytikute@yahoo.co.in (S.T.); pratikdeshmukhdpd39@gmail.com (P.D.); nachavan@yahoo.in (N.C.);; 2Maximum Containment Laboratory, ICMR-National Institute of Virology, 130, MCC, 1, Pashan—Sus Rd, Pashan, Pune 411021, Maharashtra, India; anitaaich2008@gmail.com (A.S.); hellopragya22@gmail.com (P.Y.)

**Keywords:** HFMD, enteroviruses, CV-A6, RD cell line, TEM, nonsynonymous mutation

## Abstract

Coxsackievirus-A6 (CV-A6) is responsible for more severe dermatological manifestations compared to other enteroviruses such as CV-A10, CV-A16, and EV-A71, causing HFMD in children and adults. Between 2005 and 2007, the recombinant subclade D3/RF-A started to expand globally, and a CV-A6 pandemic started. The study aimed to conduct whole-genome sequencing (WGS) of an isolated CV-A6 strain from currently circulating HFMD cases from India in 2022. Gene-specific RT-PCR and sequencing were used to perform molecular characterization of the isolated virus. Confirmation of these isolates was also performed by transmission electron microscopy and WGS. Among eleven positive clinical enterovirus specimens, eight CV-A6 strains were successfully isolated in the RD cell line. Isolates confirmed the presence of the CV-A6 strain based on VP1 and VP2 gene-specific RT-PCR. Sequences of isolates were clustered and identified as the novel CV-A6 strain of the D3/Y sub-genotype in India. The studies revealed that the D3/Y sub-genotype is being introduced into Indian circulation. The predicted putative functional loops found in VP1 of CV-A6 showed that the nucleotide sequences of the amino acid were a remarkably conserved loop prediction compatible with neutralizing linear epitopes. Therefore, this strain represents a potential candidate for vaccine development and antiviral studies.

## 1. Introduction

HFMD is a common viral disease that infects young children below the age of five. The disease can be defined as an enterovirus (EV)-associated exanthem rash characterized by a febrile illness, oropharyngeal ulcers, and vesicular rashes on the hands, feet, palms, buttocks, knees, and soles [1]. In some cases, the disease may progress to cause neurological complications such as encephalomyelitis [2]; also, it may develop to cause aseptic meningitis, acute flaccid paralysis, and brainstem encephalitis.

Since 2008, atypical HFMD outbreaks have been primarily associated with the CV-A6 variant [3,4,5,6,7]. Contemporary surveillance data have demonstrated the potential of the CV-A6 virus for epidemic spread around the world, particularly in Brazil [8], China [9], France [10], Hong Kong [11], India [12], Japan [13], and Turkey [14]. CV-A6 has been associated with severe clinical symptoms in both children and adults in several nations, including Argentina [15], Brazil [16], China [17], Israel [18], Italy [19], and Japan [20]. These symptoms include a rash that resembles vasculitis and vesiculobullous exanthema.

CV-A6 is a non-enveloped positive-sense RNA virus and belongs to the genus Enterovirus, family *Picornaviridae*. It is a 7.4 kb long genome that contains one polyprotein, which viral proteases break to create suitable proteins. Structural proteins are essential for viral entry into host cells and immune evasion; the capsid protects the viral genome and makes it easier for the virus to attach to host cells during infection [21]. CV-A6 is spread by contact with respiratory or fecal fluids or other body fluids such as saliva, nasal mucus, blister fluid, and vesicles of infected individuals. Clinical manifestations of CV-A6-associated HFMD differ from traditional HFMD caused by EV-A71 or CV-A16. Atypical herpes and onychoptosis are the most common clinical symptoms of CV-A6-associated HFMD [22]. In some cases of patients with CV-A6, the symptom of Beau’s lines occurs [23].

Due to limited data on the molecular characterization and whole-genome sequencing of CV-A6 strains from India, it is necessary to study the circulating lineages/clades of CV-A6 strains in the Indian population. By determining the circulating genotypes and recombinant strains of CV-A6 in India during the 2022 HFMD outbreak, we were able to investigate the evolutionary history of the disease. Using VP1/VP2 sequence analysis, we compared and analyzed the genetic characteristics of these recently sequenced CV-A6 variations as well as their evolutionary connections to other known worldwide strains. To better understand the prevalence of these recombinant strains, we also sequenced and characterized the almost complete genomes of eight typical CV-A6 strains circulating in India.

## 2. Materials and Methods

### 2.1. Sample Collection and Detection of EV RNA

Vesicular swabs (*n* = 225), throat swabs (*n* = 135), oral lesion swabs (*n* = 06), serum (*n* = 16), stool (*n =* 03), and urine (*n* = 40) specimens are among the 425 specimens, which were collected from symptomatic patients with HFMD from the referred hospitals and dispensaries.

All patients ranged in age from less than a year to more than twenty-one years, and their symptoms included fever; blisters on their hands, feet, and buttocks; and even mouth ulcers. One patient also had blisters in the genital region. The patient age distribution revealed that 23.9% of patients were between the ages of 11 and 35, and the male-to-female ratio was 1.92:1. Of the patients, 76.1% were under the age of 10.

Details about the patient including symptomatic conditions, type of clinical specimens, and a consent letter from the parents were obtained. All specimens were initially screened for pan-enterovirus qRT-PCR, followed by semi-nested RT-PCR targeting the VP1 region for typing. Among these characterized specimens, CV-A6-positive specimens were further selected for virus isolation based on the Ct values of qRT-PCR.

### 2.2. Virus Isolation

Throat swabs, serum, and vesicle swabs (vesicular fluid) were taken from HFMD patients. Those molecularly characterized as CV-A6 were selected for virus isolation. The procedure was carried out in a susceptible RD cell line. A total of eleven clinical specimens such as throat swabs, vesicular swabs, and serum were selected to attempt virus isolation. RD cells were prepared in Twenty-four-well plates using minimum essential medium (MEM Hi-Media, Thane, India) containing 10% fetal bovine serum, 100 µg/mL streptomycin, 100 U/mL penicillin, and a 37 °C incubation temperature with a supply of 5% CO_2_. Clinical specimens were taken as undiluted (neat) and in a 1:10 dilution and inoculated onto the RD cell line, along with cell controls. At least twice a day, the cells were observed for morphological changes. The infected cells showing cytopathic effects (CPEs) were harvested and blind passages were carried out further.

### 2.3. Virus Propagation and Estimation of TCID_50_

Among eight CV-A6 isolates passaged in the RD cell line using 24-well plates, one of the representative isolates was selected for propagation on a large scale. Scale-up was performed in T25, T75, and finally in T225 cm^2^ tissue culture flasks. TCID_50_ of the harvested isolate was further carried out in 96-well plates of the RD cell line by tenfold dilution. Cell sheet clearing indicated CPE, which represents cell death due to viruses. The Reed and Muench method was used for the calculation of titer.

### 2.4. Transmission Electron Microscopy (TEM)

Visualization of virus particles from virus isolates was acquired by transmission electron microscopy (TEM). For TEM analysis, negative staining of the sample was carried out with phosphotungstic acid as described earlier [24]. Samples were prefixed with 1% glutaraldehyde and negatively stained and a grid was examined under 100 KV in transmission electron microscopy (Tecnai 12 BioTwin ^TM^ (FEI, Eindhoven, The Netherlands)). Images were captured with a side-mounted 2 k × 2 k CCD camera (Megaview III, Olympus, Tokyo, Japan).

### 2.5. Confirmation of Enterovirus by Real-Time q-PCR from Isolates

RNA was extracted using QIAamp Viral RNA Mini Kit (Qiagen, Hilden, Germany) from the passaged viral lysate and subjected to real-time PCR (CFX96^TM^ Real-Time System, Bio-RAD, Hercules, CA, USA) to confirm enterovirus positivity. Sanger sequencing was performed to determine the sequences of nucleotide bases. Reverse transcription polymerase chain reaction (RT-PCR) was performed to amplify the whole VP1 and VP2 genes using specific primers (Table 1) [25]. Cycle sequencing PCR was carried out using a BigDye direct cycle sequencing kit using specific primers for VP1 and VP2 genes. Using the BLAST search tool (http://www.ncbi.nlm.nih.gov/blast (accessed on 3 April 2023)), the sequence identity of the EV strains was determined. Molecular Evolutionary Genetics Analysis Version 6.0 (MEGA6) provided 1000 bootstrapping replicates for the phylogenetic analysis, which was conducted using the Maximum Likelihood approach.

### 2.6. Whole-Genome Sequencing by Next-Generation Sequencing (NGS)

Whole-genome sequencing of the CV-A6 isolates was carried out using next-generation sequencing on the Illumina Miniseq platform. Using the Qubit RNA High Sensitivity (HS) kit (Thermo Fisher Scientific, Waltham, MA, USA), the concentration of extracted RNA was quantified using the Qubit^®^ 2.0 Fluorometer (Invitrogen, Life Technologies, Carlsbad, CA, USA) and then kept at −80 °C until needed. Using the NEB/Next rRNA depletion kit (New England Biolabs, Ipswich, MA, USA) (human/mouse/rat), the host’s ribosomal RNA was suppressed. Agencourt AMPure XP beads (Beckman Coulter, Brea, CA, USA) were used to further purify this RNA. The Qubit RNA High Sensitivity (HS) kit was used to quantify the depleted RNA. As recommended by the manufacturer for the depleted RNA, an RNA library was generated using the TruSeq Stranded mRNA LT Library preparation kit (llumina, San Diego, CA, USA). The Illumina Miniseq platform was loaded with the normalized library. After completion of the run, CLC Genomics. Workbench software Version 20 (CLC, Qiagen, Aarhus, Denmark) was used to import and analyze FASTQ data. A de novo method and reference mapping were both used to obtain the whole sequences of the etiological agent. Using MEGA software version 11, a Maximum Likelihood tree was constructed for the VP1 gene and the whole genome. A 1000-replication bootstrap was carried out. The reference-based assembly method in the workbench was used to retrieve the Coxsackievirus A6 sequences. OP896720.1 was used as the reference for CV-A6 strains.

Nucleotide substitution patterns and rates were estimated under the Tamura–Nei model (+G). A discrete Gamma distribution was used to model evolutionary rate differences among sites (5 categories, [+G]). The mean evolutionary rates in these categories were 0.00, 0.03, 0.20, 0.82, and 3.95 substitutions per site. The nucleotide frequencies are A = 27.81%, T/U = 24.93%, C = 23.52%, and G = 23.74%.

Each entry shows the probability of substitution (r) from one base (row) to another base (column). For simplicity, the sum of r values is made equal to 100. Rates of different transitional substitutions are shown in bold and those of transversion substitutions are shown in italics (Table 2). These evolutionary analyses were conducted in MEGA software version 11.

### 2.7. Nucleotide Sequence Accession Numbers

All newly generated complete CV-A6 genome sequences obtained in this study have GenBank accession numbers from OR734733 to OR734740.

## 3. Results

### 3.1. Molecular Characterization of Enterovirus

A total of 425 clinical samples from 196 patients who had been diagnosed with HFMD were screened for pan-enterovirus using qRT-PCR. By using qRT-PCR, only 54.6% [107/196] of these patients showed evidence of EV RNA. Out of the patients who tested positive for human enterovirus, 42.0% [45/107] tested positive for CV-A16 and 28.97% [31/107] tested positive for CV-A6.

### 3.2. Virus Isolation

Based on Ct values (<25), eleven CV-A6-positive specimens (in duplicates) were inoculated onto a confluent monolayer of rhabdomyosarcoma (RD) cells. The infected cell lines were monitored daily for 5 to 6 days. The cells showed cytopathic effect (CPE) on day 4 postinfection. After 24 h of incubation, the cells begin to show morphological changes (Figure 1B). Virus-infected cells were harvested after showing CPE of 90 to 95% (Figure 1D). Another freeze–thaw process was performed and the inoculum was used for the next passage. The cells showed morphological changes from passage stage 3 (P-3) and a further blank passage was performed until passage stage 6 (P-6) and confirmed for the presence of CV-A6 using the VP1 and VP2 gene-specific RT-PCR cycle sequencing. Among eleven CV-A6-positive specimens, eight CV-A6 strains were successfully isolated in RD cell lines. A TCID_50_ assay was performed on RD cells to estimate the viral titer of CV-A6. Cell sheet clearing indicates CPE, which represents cell death due to viral infectivity. From the experiment, 50% of wells showed CPE at the 10^−10^ dilution, which implies the titer is 10^10^.

### 3.3. Confirmation of Virus Particles from Isolate by Transmission Electron Microscopy

TEM imaging of cleared and negatively stained isolates from individuals with CV-A6 (*n* = 05) revealed the various virus-like particles belonging to the *picornaviridae* family with a diameter of 30 nm (Figure 2). All HFMD patients were between 1 and 3.6 years old and had rashes on their hands and feet. Fever and mouth ulceration have been reported.

### 3.4. Confirmation of Isolates by q-Real-Time PCR and VP-1/VP-2 Sequencing

The data and Ct (cycle threshold) values generated from the CFX96^TM^ Real-Time System are arranged in ascending order. Ct values of CV-A6 isolates (783, 035, 722, 726, 273, 051, 064, 087, 274, and 085) were obtained by performing a real-time PCR assay for enterovirus and viral passages were confirmed. The Ct value of the isolates ranged between 11 and 27. Analysis of obtained data was further used for whole-genome sequencing experiments.

Molecular studies were conducted for the identification of CV-A6 strains and comparative analysis of the confirmed CV-A6 strains with other CV-A6 strains. Virus serotypes were identified using the BLASTn program from GenBank. The VP1 region of th confirmed strains CV-A6/087, CV-A6/051, CV-A6/722, CV-A6/726, CV-A6/274, and CV-A6/273 showed percentages of identity of 100%, 98.04%, 96.54%, 92.76%, 78%, 68.79% with OK635720.1, LC481419.1, OK635720.1, OP896720.1, MN233825.1, and OK635720.1, respectively. The VP2 region of the confirmed strains CV-A6/087, CV-A6/035, CV-A6/064, CV-A6/274, CV-A6/051, CV-A6/273, and CV-A6/783 showed percentages of identity of 98.19%, 97.46%, 96.52%, 96.04%, 95.27%, 94.41%, 85.80% with OP896719.1, OP896719.1, OP896719.1, MZ491032.1, MN845889.1, OL830040.1, and MN845859.1, respectively.

### 3.5. Whole-Genome Sequencing by Next-Generation Sequencing (NGS)

To obtain new insights into the genetic diversity of CV-A6, we sequenced the whole genomes of eight isolates of CV-A6 strains with good CPE that were collected in 2022 (Table 3). All eight isolate sequences were derived from eight different patients. These sequences were then compared to GenBank reference sequences. The CV-A6 strains in this study had a nearly complete genome of 7435 nt, which included an ORF encoding a polyprotein precursor with 2201 amino acids and a portion of the 5′ UTR.

The whole genome of study isolates was aligned along with full whole-genome sequences of reference strains (OP896720, MN845848, OP896719, KM114057, MH780756, AB779614, JN203517, AY421764, JQ364886, KP143074, MN845781, MT814422, OL830039). All eight CV-A6 strains grouped together to form a clade, which was confirmed by the 5′ UTR, P1, and P2 regions having high bootstrap values (96–100%). Within the structural protein of the investigated strains, a strain belonging to subclade D3 that was identified in Thailand in 2022 had the most significant identity (97.7–99.5% nt identity). To elucidate the mutation in the VP1 gene (914 bp) of the D3/Y strains of CV-A6 compared the amino acid and nucleotide variation between the VP1 sequences of representative isolated strains and reference D3/Y strains, sequence alignment was performed and analyzed with reference CV-A6 strain (OP896720.1) by Needleman–Wunsch alignment and MEGA software version 11.

All CV-A6 isolates showed 97% to 98% identity with the reference strain with 2% to 3% nucleotide substitution in the VP1 region. The VP1-encoding gene of EVs is crucial in preventing the host immune response from being triggered. This study found 33 sites of nonsynonymous mutations in the VP1 gene of the isolates. Among them, 2456 (Q 578 P), 2465 (V 581 A), 2531 (L 603 P), 2657 (R 656 Q), 2666 (N 648 S), 2783 (V 687 A), 2813 (L 697 S), 2858 (R 712 H), 2870 (C 716 Y), 2888 (Y 722 C), 2897 (G 725 E), 2900 (L 726 P), 2906 (N 728 S), 2975 (I 751 T), 3077 (I 785 T), 3083 (N 787 S), 3164 (M 814 T), 3218 (S 832 F), 3221 (A 833 D), 3230 (L 836 P), 3272 (P 850 L), and 3290 (V 856 A) were present and more frequently seen in the isolates as compared to the reference strain from the D3/Y subclade (Appendix A).

The VP1 region sequences were categorized into clades A–D, and clade D was further subdivided into the D1–D3 subclades based on established classification criteria. This study found 33 types of non-synonymous mutations in the VP1 gene of the isolates. Based on the whole genomes of CV-A6 isolates and other reference strains, a phylogenetic tree was built using NGS data. Every significant functional area of the genome was analyzed separately. The evolutionary history was determined by the Neighbor-Joining method. The percentage of replicate trees with clustered taxa in the bootstrap test (1000 repetitions) is indicated above the branches (Figure 3). With branch lengths matching the evolutionary distances used to build the phylogenetic tree. The evolutionary distances are reported in base substitutions per site and have been derived using the Maximum Composite Likelihood method. The final dataset comprised 7439 locations. MEGA11 was used to carry out evolutionary analysis, with branch lengths matching the evolutionary distances used to build the phylogenetic tree. The evolutionary distances are reported in base substitutions per site and have been derived using the Maximum Composite Likelihood method.

Sequences of the isolates were clustered and identified to the new D3/Y sub-genotype, which exist in the dominant CV-A6 genotype that is currently in circulation in Thailand [26]. Most of the global outbreaks that have occurred were due to sub-genotype D3. The findings suggest that the persistent international circulation of CV-A6 may be due to the high transmission, infectivity, and virulence of sub-genotype D3 strains [27]. It can be suspected that the D3/Y sub-genotype has now been introduced into Indian circulation.

## 4. Discussion

Infection due to CV-A6 causes HFMD in both children and adults. Despite the major risk that CV-A6 illnesses represent to public health, our knowledge of the mechanisms by which new CV-A6 strains originate remains restricted [26]. Since 2008, CV-A6 strains have become the main genotype responsible for global epidemics of atypical HFMD [3]. Several sporadic cases of HFMD with high morbidity and death have occurred, mainly in Southeast Asian nations since 1997 and India after 2003 [23]. Multiple enterovirus serotypes were found to circulate simultaneously in HFMD patients from Bhubaneswar, Odisha, in 2009, southern and eastern India in 2009–2010 [22], and northern Kerala in 2015–2016. CV-A6 was the leading cause of HFMD in India from 2015 to 2017, followed by CV-A16 and CV-A10 [28]. Co-circulation of CV-A6 and CV-A16 was reported to cause HFMD in Mumbai in May–June 2018 [29]. It is essential to study the CV-A6 strains causing HFMD that are currently circulating in India. Due to limited data on the isolation of recent CV-A6 strains, its molecular characterizations and whole-genome sequencing will emphasize the genetic changes that occurred during this period. This study aimed to isolate, characterize, and perform whole-genome sequencing of the viral isolates causing HFMD in India. The study was based on three different research parameters, which comprise the isolation of the viral agents causing HFMD in the RD cell line. Molecular studies were conducted for identification, comparative analysis, and whole-genome sequencing by using next-generation sequencing using the Illumina platform for comparative studies. It provides knowledge of the genetic organization of the currently circulating CV-A6 viral strains. The study also provides information on the molecular characterizations of etiological agents and the evolutionary pattern of different CV-A6 isolates from various geographical locations in India.

Isolation of the clinically positive specimens was carried out in susceptible RD cell lines. The cell line was infected with the positive enterovirus samples. After 4 to 5 days of incubation, the cell line showed morphological changes in the form of cytopathic effect (CPE). Viral-infected cells with 90% CPE were collected, freeze–thawed, and utilized as an inoculum for subsequent passages. The cells showed morphological changes from passage level 3 (P-3) and further blind passaging was performed till passage level 6 (P-6) to get the best viral titer and confirm the presence of CV-A6 based on VP1 and VP2 gene-specific RT-PCR and cycle sequencing. Among eleven positive enterovirus specimens, eight CV-A6 strains were successfully isolated in RD cell lines. The representative viral agent was propagated on a large scale and further processed for virus titration. Virus titration was performed using tenfold dilutions of viral lysate and the titer was estimated in TCID_50_. Isolated virus showed a 50% infectivity dose at a 10^10^ titer. VP1 and VP2 typing enables us to determine the sub-genotype of the viral strain. Sequences of isolates were clustered and identified to the new CV-A6 D3/Y sub-genotype, which existed in the dominant CV-A6 genotype circulating in Thailand [27]. Most of the global outbreaks that occurred thereafter were due to sub-genotype D3. The data show that ongoing international circulation of CV-A6 might be attributed to the high transmission, infectivity, and virulence of sub-genotype D3 viruses. It is suspected that the D3/Y sub-genotype is now emerging in India.

The phylogenetic tree was constructed using NGS data, and the whole genome of the study isolates was aligned along with whole genome sequences of reference strains. This study provided essential data, which will be useful for further studies. However, the VP1 gene is regarded as the most informative and sturdy region in evolutionary studies because of its wide range of diversity and lack of involvement in recombination. [29]. Human enteroviruses evolve by genetic drift and, over considerably longer periods, antigenic diversity in the structural gene region encoding the viral capsid, which includes VP1 [30]. When comparing strains with the reference strain (OP896720.1), nucleotide alterations and amino acid substitutions in the VP1 gene (914 bp) of representative isolates (CV-A6/087, CV-A6/274, CV-A6/051, CV-A6/783) were observed from the same subclade of isolates. All isolates showed 97% to 98% identity with the reference strain (OP896720.1) with 2% to 3% nucleotide substitution in the VP1 region. The VP1-encoding gene of EVs is crucial in preventing the host immune response from being triggered. This study found 33 sites of nonsynonymous mutations in the VP1 gene of the isolates. Among them, 22 mutations were at high frequencies in the isolates as compared to the reference strain (OP896720.1) from the D3/Y sub-genotype.

In previous studies, the amino acid changes between the four virus strains and the subclade D3/A strains from Thailand were compared to mutations in the subclade D3/Y strains from Thailand. There were fewer nonsynonymous mutations in the structural proteins (at the fifth position) than in the non-structural proteins. A crucial part of EVs’ ability to elude the host immune response is their VP1-encoding gene. The VP1 gene of the sub-genotype CV-A6 D3/Y strains showed nine locations with nonsynonymous alterations. Among these, three mutations were present more frequently in the D3/Y sub-genotype [26]. The whole genome was analyzed using previous CV-A6 reference strains. Sequences of the isolates were clustered and identified to the new CV-A6 D3/Y sub-genotype, which was the predominant CV-A6 genotype circulating in Thailand. It can be suspected that the CV-A6 D3/Y sub-genotype is now in Indian circulation. Analysis of the complete VP1 gene revealed multiple nucleotide changes and amino acid substitutions and found 33 sites of nonsynonymous mutations in the VP1 gene of the isolates. Studies have shown that multiple protruding loops, such as the BC loop (residues 97–105), the EF loop (residues 163–177), and the GH loop (residues 208–225), and proximity of the C-terminus (residues 253–267) in VP1 have been identified as the significant antigenic proteins exposed on the viral surface. The host cell receptor for CV-A6 was found to be Kringle-Containing Transmembrane Protein 1 (KREMEN1). The BC, DE, EF, and HI loops on the surface of VP1 are the preferred binding sites for many short RNA viral receptors. Mutations in the 712th, 716th, 722nd, 725th, 751st, 785th, and 833^rd^ amino acid positions of the VP1 region can change the structural characteristics of the capsid and alter the virus’ ability to bind to the KREMEN1. Further studies can be conducted on how these amino acids alter the structural changes in the viral capsid and whether they help to bind the KREMEN1 receptor and the role of amino acids in causing infectivity. Predicted putative functional loops found in the VP1 of CV-A6 showed that the nucleotide sequences of the amino acid were remarkably conserved loops; this prediction is compatible with neutralizing linear epitopes. As a result, the D3/Y strains further represent a potential candidate for the development of a CV-A6 vaccine and antiviral studies.

## 5. Conclusions

This study provides further information on the molecular aspects and evolutionary patterns of different CV-A6 isolates from different geographical regions, revealing the importance of evolutionary relationships, as well as the genetic organization of the current circulating CV-A6 viral strains. The data will facilitate the development of effective diagnostic tools, antiviral therapies, and vaccines for the prevention and control of CV-A6-associated HFMD.

## Figures and Tables

**Figure 1 microorganisms-12-00490-f001:**
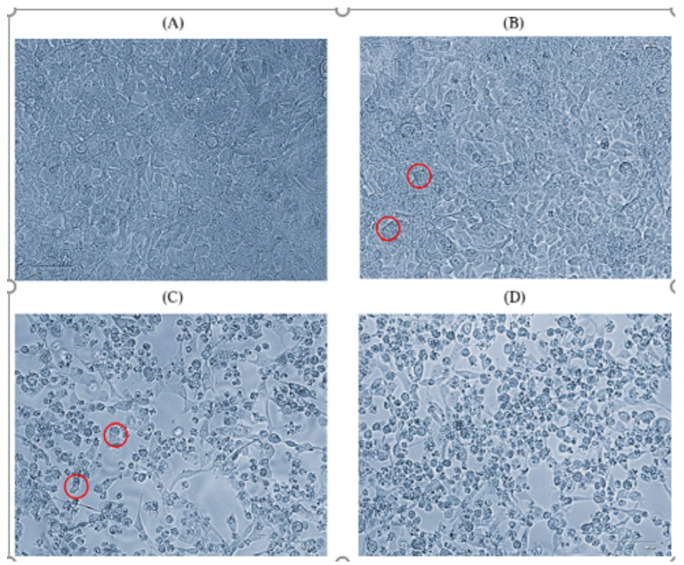
Microscopic images (40×) of RD cell line on day 4 of passage level 6. (**A**) Normal cells, (**B**) initiation of CPE, (**C**) 70% CPE, (**D**) 90% CPE (magnification 40×, EVOS XL core). Red circle indicates the CPE.

**Figure 2 microorganisms-12-00490-f002:**
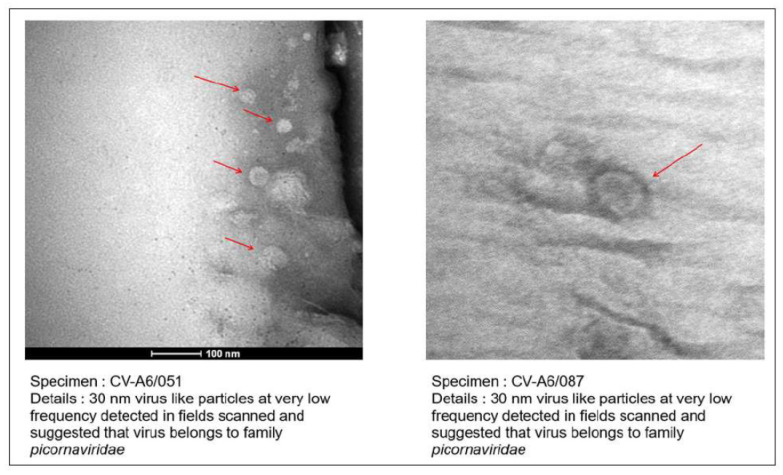
TEM imaging from the virus isolates from HFMD patients. Red arrows showing the 30 nm virus like particles.

**Figure 3 microorganisms-12-00490-f003:**
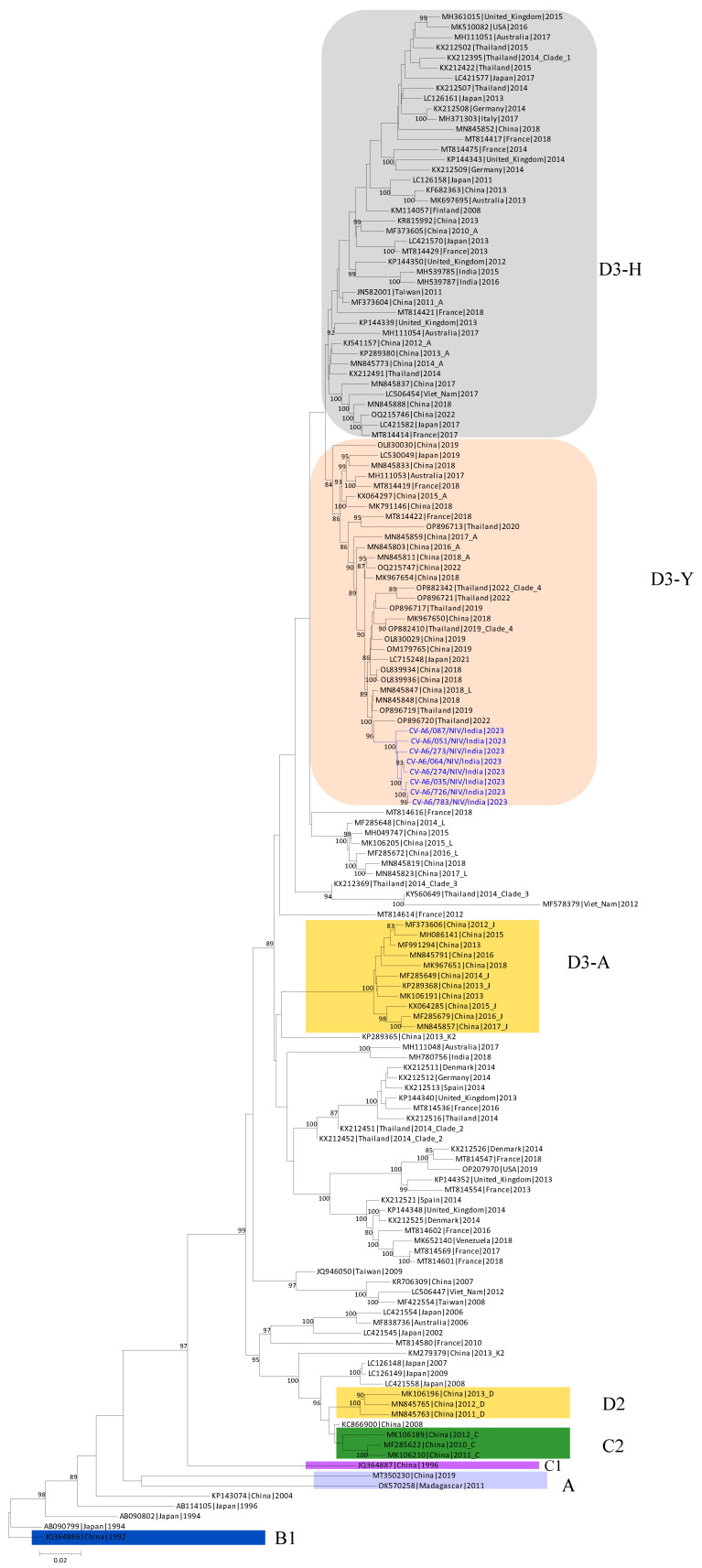
Time-scaled phylogenetic tree of sequences of coxsackievirus (CV)-A6 variants. The phylogenetic tree was generated using CV-A6 sequences characterized in this study (blue) and previously published sequences. Clades described in previous studies (A, B1, C1, C2, D2, and D3) are shown.

**Table 1 microorganisms-12-00490-t001:** Primers used for VP1 and VP2 gene-specific RT-PCR and sequencing.

Primers	Sequences (5′–3′)	Region	Nucleotide Positions (Range)
224 Forward	GCIATGYTIGGIACICAYRT	VP1	1977–1996
222 Reverse	CICCIGGIGGIAYRWACAT	VP1	2969–2951
89 Forward	CCAGCACTGACAGCAGYNGARAYNGG	VP1	2602–2627
88 Reverse	TACTGGACCACCTGGNGGNAYRWACAT	VP1	2977–2951
12 Forward	ATGTAYGTICCICCIGGIGG	VP2	2917–2936
22 Forward	GCICCIGAYTGITGICCRAA	VP2	3408–3389
32 Reverse	GTYTGCCA	VP2	3009–3002

**Table 2 microorganisms-12-00490-t002:** The pattern of nucleotide substitution.

	A	T	C	G
**A**	-	*2.12*	*2.1*	**15.01**
**T**	*2.44*	-	**24.91**	*2.1*
**C**	*2.44*	**25.14**	-	*2.1*
**G**	**17.43**	*2.12*	2.1	-

**Table 3 microorganisms-12-00490-t003:** Data was obtained from NGS with the percentage of identity and genome coverage as well as genomic length of all CV-A6 isolates. All isolates were mapped with OP896720.1 as aligned in the NCBI server, and it is taken as a reference strain.

Isolate ID	Percentage of Identity	Reference Strain	Consensus Length	Genome Coverage
CV-A6/726/NIV/IND/2023	98.25%	CV-A6 Isolate Thailand/2022 (OP896720.1)	7435	99%
CV-A6/274/NIV/IND/2023	98.20%		7435	99%
CV-A6/064/NIV/IND/2023	98.12%		7435	100%
CV-A6/035/NIV/IND/2023	97.87%		7435	100%
CV-A6/087/NIV/IND/2023	97.85%		7435	100%
CV-A6/783/NIV/IND/2023	97.82%		7435	100%
CV-A6/723/NIV/IND/2023	97.74%		7435	100%
CV-A6/051/NIV/IND/2023	97.00%		7435	99%

## Data Availability

Data are contained within the article and Appendix A.

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
