# Peer review of "Emergence of Recombinant Subclade D3/Y in Coxsackievirus A6 Strains in Hand-Foot-and-Mouth Disease (HFMD) Outbreak in India, 2022"

_microorganisms, 2024, doi:10.3390/microorganisms12030490_

Round 1

Reviewer 1 Report

Comments and Suggestions for Authors

In this manuscript, the authors collected 425 specimens to identify evolutionary of CV-A6 in India during 2022 HFMD outbreak. the authors compared VP1/VP2 sequence of CV-A6 variants and their phylogenetic relationships with reported strains. I think the manuscript can be published in Microorganisms with the following modifications.

1. Only 10 virus strains were isolated from 425 samples, and 6 VP1 sequences and 7 VP2 sequences were amplified for analysis, lacking representativeness.

2. In Section 3.5, it is difficult to understand how 6 VP1 sequences and 7 VP2 sequences are compared with the reported sequences.

3. In Section 3.6, where is Table1?

4. Figure 3 is unclear

Author Response

In this manuscript, the authors collected 425 specimens to identify evolutionary of CV-A6 in India during 2022 HFMD outbreak. the authors compared VP1/VP2 sequence of CV-A6 variants and their phylogenetic relationships with reported strains. I think the manuscript can be published in Microorganisms with the following modifications.

1. Only 10 virus strains were isolated from 425 samples, and 6 VP1 sequences and 7 VP2 sequences were amplified for analysis, lacking representativeness.

Reply: Among these 425 HFMD samples, only 31 were found CVA6. Based on Ct values (<25), eleven CV-A6 positive specimens (in duplicates) were inoculated onto a confluent monolayer of rhabdomyosarcoma (RD) cells. These 11 molecularly characterized clinical samples were further processed for isolation. We found only eight isolates with good CPE. This part we have already described in the section 3.2

2. In Section 3.5, it is difficult to understand how 6 VP1 sequences and 7 VP2 sequences are compared with the reported sequences.

      Reply: We agree with the comment of Reviewer. Modification included in the section 3.5 as per suggestion.

3. In Section 3.6, where is Table1?

      Reply: Included in the revised manuscript section 2.5 of Material and Methods

4. Figure 3 is unclear

      Reply:  Modified as per suggestion and attached separately after the references. We can provide separately in jpeg if required.

Reviewer 2 Report

Comments and Suggestions for Authors

This study demonstrates that the coxsackievirus A6 (CV-A6) genotype that was previously identified with severe cases of hand foot and mouth disease (HFMD) in Thailand was also found in cases in India.  Using 425 clinical specimens from 196 patients with HFMD, the authors identified 11 positive for CV-A6 viral RNA.  These specimens which generated cytopathic effect in RD cells (8) were then amplified in cell culture and a nearly complete genome sequence was generated from virus RNA using next generation sequencing.  Phylogenetic comparison with CV-A6 sequences available from databases which identified these as belonging to the same subgenotype, clade D3/Y, as the Thailand HFMD CV-A6 isolates.  The authors analyzed the amino acid sequence of these isolates and identified VP1 sites at which these isolates had variation from the sequence of the CV-A6 D3/Y isolate from a HFMD case in Thailand in 2022. 

This does provide nearly complete CV-A6 sequences from Indian cases of FMDV in 2022 and demonstrates that they are in the subgenotype D3/Y.  It would be instructive to provide the analysis of the VP1 variation in the 8 isolates that is mentioned in the discussion.  The variation in 4 isolates is provided in a supplementary table but only identifies the site, nucleotide and amino acid.  It would be interesting to know what part of the CV-A6 VP1 protein these sites encoded and whether these were variations found in other isolates of CV-A6 and how frequently.  In other words, were these sites in positions which allowed variation, and what amino acids were present at these sites?

The resolution of figure 3 is very poor, even in the version given outside of the manuscript.

Were the 8 sequences derived from 8 different patients?  I assume that given there are 4 different genomic sequences, that there were at least 4 different patients.  This should be clearly stated. On lines 180-182, the authors refer to a Table 1 that is not present in the manuscript with information on the 8 sequenced isolates.  I think this cannot be the VP1 variation table (Supplementary Table 1) which has only sequence information on 4 isolates. 

Section 3.3 is a repeat of information in the methods section.  The first sentence (line 154: “TCID50 assay performed on RD cells to estimate the viral titer of CV-A6.”) should be the last sentence in section 3.2.

The authors state on lines 107-109: “Cycle sequencing PCR was carried out using BigDye direct cycle sequencing Kit using specific primers for VP1 and VP2 genes.” The specific primers used should be identified in methods or a reference should be given.

This is a description of the generation of sequences of CV-A6 important in recent outbreaks of HFMD in India and adds to the extent of information available on CV-A6.  There is information available in the analysis of the coding sequences of these isolates which is referred to but not shown.  It would add to the significance of this paper to provide this type of information.   

Comments on the Quality of English Language

On line 50, onchoptosis should be spelled onychoptosis.

Line 51-52: “In some cases of CV-A6 infected patients experience the Beau’s lines.” could be better as “ In some cases of CV-A6 infected patients the symptom, Beau’s lines, occur.”

Line 78: “The molecularly characterized as CV-A6” should be “Those molecularly characterized as CV-A6”.

Lines 192-194: “To elucidate the mutation in VP1 gene (914 bp) of the D3/Y strains of CV-A6 compared the amino acid and nucleotide changes between VP1 sequences of representative isolated strains and reference D3/Y strain.” should be “To elucidate the mutations in the VP1 gene (914 bp) of the D3/Y strains of CV-A6 we compared the amino acid and nucleotide variation between VP1 sequences of representative isolated strains and reference D3/Y strain.”

Line 198: “VP1-encoding gene of EVs” should be “the VP1-encoding gene of EVs”

Lines 199-200: “This study found 33 types of non-synonymous mutations in the VP1 gene of the isolates.” Types should be sites.

Lines 295-297: “Analysis of complete VP1 gene revealed several nucleotide changes and amino acid substitutions and founds 33 types of non-synonymous mutations in the VP1 gene of the isolates.” Founds 33 types should be found 33 sites.

Author Response

Comment: This study demonstrates that the coxsackievirus A6 (CV-A6) genotype that was previously identified with severe cases of hand foot and mouth disease (HFMD) in Thailand was also found in cases in India.  Using 425 clinical specimens from 196 patients with HFMD, the authors identified 11 positives for CV-A6 viral RNA.  These specimens which generated cytopathic effect in RD cells (8) were then amplified in cell culture and a nearly complete genome sequence was generated from virus RNA using next generation sequencing.  Phylogenetic comparison with CV-A6 sequences available from databases which identified these as belonging to the same subgenotype, clade D3/Y, as the Thailand HFMD CV-A6 isolates.  The authors analyzed the amino acid sequence of these isolates and identified VP1 sites at which these isolates had variation from the sequence of the CV-A6 D3/Y isolate from a HFMD case in Thailand in 2022.

 This does provide nearly complete CV-A6 sequences from Indian cases of FMDV in 2022 and demonstrates that they are in the subgenotype D3/Y.  It would be instructive to provide the analysis of the VP1 variation in the 8 isolates that is mentioned in the discussion.  The variation in 4 isolates is provided in a supplementary table but only identifies the site, nucleotide, and amino acid.  It would be interesting to know what part of the CV-A6 VP1 protein these sites encoded and whether these were variations found in other isolates of CV-A6 and how frequently.  In other words, were these sites in positions which allowed variation, and what amino acids were present at these sites?  

Reply: As per reviewer suggestion we have included the detailed of these SNPs occurred in our isolates.

Included in result section 3.5 : To elucidate the mutation in VP1 gene (914 bp) of the D3/Y strains of CV-A6, we compared the amino acid and nucleotide changes between VP1 sequences of four representative isolated strains and reference D3/Y strain. Sequence alignment was done and analyzed with reference CV-A6 strain (OP896720.1) by Needleman-Wunsch alignment and MEGA software version 11. All CV-A6 isolates showed 97% to 98% identity with the reference strain with 2% to 3% nucleotide substitution in VP1 region. This study found 33 types of non-synonymous mutations in the VP1 gene of the isolates. Among them, 2456 (Q 578 P), 2465 (V 581 A), 2531 (L 603 P), 2657 (R 656 Q), 2666 (N 648 S), 2783 (V 687 A), 2813 (L 697 S), 2858 (R 712 H), 2870 (C 716 Y), 2888 (Y 722 C), 2897 (G 725 E), 2900 (L 726 P), 2906 (N 728 S), 2975 (I 751 T), 3077 (I 785 T), 3083 (N 787 S), 3164 (M 814 T), 3218 (S 832 F), 3221 (A 833 D), 3230 (L 836 P), 3272 (P 850 L), 3290 (V 856 A) were present at high frequencies in the isolates as compared to reference strain from D3/Y subclade.

Included in the section 2.5: Substitution pattern and rates were estimated under the Tamura-Nei model (+G). A discrete Gamma distribution was used to model evolutionary rate differences among sites (5 categories, [+G]). Mean evolutionary rates in these categories were 0.00, 0.03, 0.20, 0.82, 3.95 substitutions per site. The nucleotide frequencies are A = 27.81%, T/U = 24.93%, C = 23.52%, and G = 23.74%.

Each entry shows the probability of substitution (r) from one base (row) to another base (column). For simplicity, the sum of r values is made equal to 100. Rates of different transitional substitutions are shown in bold and those of transversion substitutions are shown in italics (Figure). The nucleotide frequencies are 27.82% (A), 24.22% (T/U), 24.00% (C), and 23.95% (G). Evolutionary analyses were conducted in MEGA software version 11.

Included in the Discussion last para:

Studies have shown that multiple protruding loops, such as B–C loop (residues 97–105), E–F loop (residues 163–177), G-H loop (residues 208–225), and proximity of C-terminus (residues 253–267) in VP1 have been identified as the significant antigenic proteins exposed on the viral surface. Host cell receptor for CV-A6 was found to be Kringle Containing Transmembrane Protein 1 (KREMEN1). The BC, DE, EF, and HI loops on the surface of VP1 are the preferred binding sites for many short RNA viral receptors. Mutation in position 712th, 716th, 722nd, 725th, 751st, 785th, 833rd of amino acid of VP1 region can change the structural characteristics of capsid and alter the virus ability to bind to the KREMEN1. Further studies can be conducted on how does this amino acid alters the structural changes in the viral capsid and whether it helps to bind KREMEN1 receptor and the role of amino acid in causing infectivity. The data will facilitate the development of effective diagnostic tools, antiviral therapies and to develop vaccine for the control and prevention CV-A6- associated HFMD.

Comment: The resolution of figure 3 is very poor, even in the version given outside of the manuscript.

Reply: Included in revised manuscript.

Comment: Were the eight sequences derived from eight different patients?  I assume that given there are four different genomic sequences, that there were at least four different patients.  This should be clearly stated.

Reply: As per reviewer’s suggestions, we have included the statement in the result section 3.5: “All eight isolates sequences were derived from eight different patients”.

Comment: On lines 180-182, the authors refer to a Table 1 that is not present in the manuscript with information on the 8 sequenced isolates.  I think this cannot be the VP1 variation table (Supplementary Table 1) which has only sequence information on 4 isolates. 

Reply: We agree with the reviewer’s suggestion. Table 1 is the primers sequences and Table 2 is with the NGS data of all the eight isolates. Included in the revised manuscript result section 3.5. The VP1 variation table (Supplementary Table 1) which has sequence information on 4 representative isolates, based on their ct values. 

Comment: Section 3.3 is a repeat of information in the methods section.  The first sentence (line 154: “TCID50 assay performed on RD cells to estimate the viral titer of CV-A6.”) should be the last sentence in section 3.2.

Reply: Revision has been done as per suggestion.

Comment: The authors state on lines 107-109: “Cycle sequencing PCR was carried out using BigDye direct cycle sequencing Kit using specific primers for VP1 and VP2 genes.” The specific primers used should be identified in methods or a reference should be given.

Reply: Revision has been done as per suggestion. A new table added as Primers used for VP1 and VP2 gene specific RT-PCR and sequencing  as Table 1.

Comment: This is a description of the generation of sequences of CV-A6 important in recent outbreaks of HFMD in India and adds to the extent of information available on CV-A6.  There is information available in the analysis of the coding sequences of these isolates which is referred to but not shown. It would add to the significance of this paper to provide this type of information. 

Reply: Updates has been included in discussion section.

Comments on the Quality of English Language

On line 50, onchoptosis should be spelled onychoptosis.

Reply: Corrected as per suggestion

Line 51-52: “In some cases of CV-A6 infected patients experience the Beau’s lines.” could be better as “In some cases of CV-A6 infected patients the symptom, Beau’s lines, occur.”

Reply: Corrected as per suggestion

Line 78: “The molecularly characterized as CV-A6” should be “Those molecularly characterized as CV-A6”.

Reply: Corrected as per suggestion

Lines 192-194: “To elucidate the mutation in VP1 gene (914 bp) of the D3/Y strains of CV-A6 compared the amino acid and nucleotide changes between VP1 sequences of representative isolated strains and reference D3/Y strain.” should be “To elucidate the mutations in the VP1 gene (914 bp) of the D3/Y strains of CV-A6 we compared the amino acid and nucleotide variation between VP1 sequences of representative isolated strains and reference D3/Y strain.”

Reply: Corrected as per suggestion

Line 198: “VP1-encoding gene of EVs” should be “the VP1-encoding gene of EVs”

Reply: Corrected as per suggestion

Lines 199-200: “This study found 33 types of non-synonymous mutations in the VP1 gene of the isolates.” Types should be sites.

Reply: Corrected as per suggestion

Lines 295-297: “Analysis of complete VP1 gene revealed several nucleotide changes and amino acid substitutions and founds 33 types of non-synonymous mutations in the VP1 gene of the isolates.” Founds 33 types should be found 33 sites.

Reply: Corrected as per suggestion
